# SV4D: Dynamic 3D Content Generation with Multi-Frame and Multi-View Consistency

**Yiming Xie**[1,2*]    **Chun-Han Yao**[1*]    **Vikram Voleti**[1]    **Huaizu Jiang**[2†]    **Varun Jampani**[1†]

[1] Stability AI    [2] Northeastern University

[*] Equal contribution    [†] Equal advising

## Abstract

We present Stable Video 4D (SV4D) — a latent video diffusion model for multi-frame and multi-view consistent dynamic 3D content generation. Unlike previous methods that rely on separately trained generative models for video generation and novel view synthesis, we design a unified diffusion model to generate novel view videos of dynamic 3D objects. Specifically, given a monocular reference video, SV4D generates novel views for each video frame that are temporally consistent. We then use the generated novel view videos to optimize an implicit 4D representation (dynamic NeRF) efficiently, without the need for cumbersome SDS-based optimization used in most prior works. To train our unified novel view video generation model, we curate a dynamic 3D object dataset from the existing Objaverse dataset. Extensive experimental results on multiple datasets and user studies demonstrate SV4D's state-of-the-art performance on novel-view video synthesis as well as 4D generation compared to prior works. Project page: https://sv4d.github.io.

## 1 Introduction

The 3D world we live in is dynamic in nature with moving people, playing pets, bouncing balls, waving flags, etc. Dynamic 3D object generation, also known as 4D generation, is the task of generating not just the 3D shape and appearance (texture) of a 3D object, but also its motion in 3D space. In this work, we tackle the problem of generating a 4D (dynamic 3D) object from a single monocular video of that object. 4D generation enables the effortless creation of realistic visual experiences, such as for video games, movies, AR/VR, etc.

4D generation from a single video is highly challenging as this involves simultaneously reasoning both the object appearance and motion at unseen camera views around the object. This is also an ill-posed problem as a multitude of 4D results can plausibly explain a single given video. There are two main technical challenges in training a robust 4D generative model that can generalize to different object types and motions. First, there exists no large-scale dataset with 4D objects to train a robust generative model. Second, the higher dimensional nature of the problem requires a large number of parameters to represent the 3D shape, appearance, and motion of an object.

As a result, several recent techniques (Singer et al., 2023b; Bahmani et al., 2024; Ling et al., 2024; Ren et al., 2023; Zhao et al., 2023; Jiang et al., 2024c; Zeng et al., 2024; Yin et al., 2023) optimize 4D content by leveraging priors in pre-trained video and multi-view generative models via score-distillation sampling (SDS) loss (Poole et al., 2023) and its variants. However, they tend to produce unsatisfactory results due to independent modeling of object motion using video models, and novel view synthesis using multi-view generative models. In addition, they tend to take hours to generate a single 4D object due to time-consuming SDS-based optimization. Concurrent works (Yang et al., 2025; Sun et al., 2025) try to partially address these issues by jointly sampling novel view videos (along both video frame and view axes) using both video and multi-view generative models; and then using the resulting novel view videos for 4D optimization. This results in a better novel view video synthesis with multi-view dynamic consistency, but several inconsistencies still remain due to the use of separate video and multi-view generative models.

In this work, we propose Stable Video 4D (SV4D) model that takes as input a single video, such as of a dynamic object, along with a user-specified camera trajectory around the object, and outputs

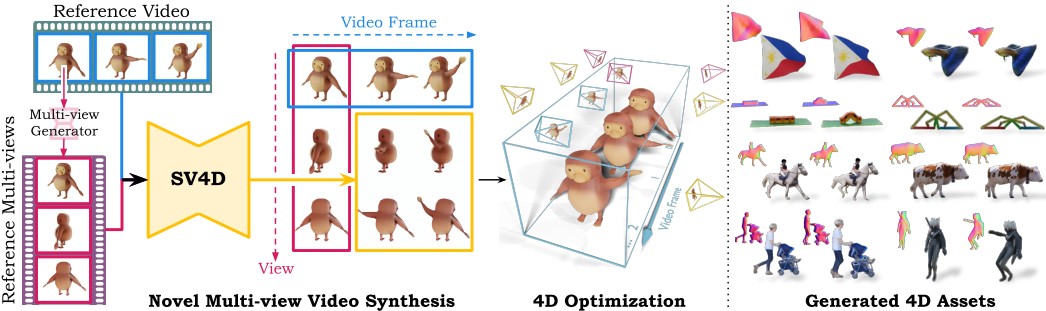

Figure 1: **Stable Video 4D (SV4D)** framework overview and generated 4D assets. We adapt and train a video diffusion model to generate novel view videos, conditioned on a single-view video and a multi-view orbital video of the first frame. SV4D generated novel view videos are consistent across both view and motion axis, which we directly use to optimize dynamic 3D objects without the cumbersome SDS loss.

videos of the object along each of the specified camera views. That is, given a video with $F$ number of frames and a camera trajectory with $V$ number of camera views, SV4D outputs a $V \times F$ grid of images, as illustrated in Fig. 1. Given a reference video, we first obtain the reference multi-views for the first frame using an off-the-shelf multi-view generator (SV3D (Voleti et al., 2024)). SV4D then jointly outputs the remaining grid of images as highlighted by a yellow box in Fig. 1. In contrast to prior and concurrent works, SV4D jointly reasons along both view and motion axes, resulting in state-of-the-art multi-frame and multi-view consistency in the output novel view videos.

Specifically, we start with Stable Video Diffusion (SVD) (Blattmann et al., 2023a), a state-of-the-art video generator, and equip it with two attention blocks: view attention and frame attention. The view attention block aligns the multi-view images at each video frame, conditioning on the first view *i.e*, in the reference video. Similarly, the frame attention block aligns the multi-frame images at each view, conditioning on the first frame at each view, *i.e*, reference multi-view. This dual-attention design leads to significantly improved dynamic and multi-view consistency compared to prior art.

A key challenge here is that SV4D needs to simultaneously generate the $V \times F$ grid of images, which can quickly become large with long input videos; making it infeasible to fit into memory even on modern GPUs. As a remedy, we propose techniques to sequentially process an interleaved subset of input frames while also retaining consistency in the output image grid. After generating the multiple novel view videos, we optimize a 4D representation of the dynamic 3D asset, as illustrated in Fig. 1.

Given the lack of large-scale 4D datasets, we carefully initialized SV4D weights with those from SVD (Blattmann et al., 2023a) and SV3D (Voleti et al., 2024) networks, thereby leveraging the priors learned in the existing video and multi-view diffusion models. To further train SV4D, we carefully curated a subject of Objaverse (Deitke et al., 2023b;a) dataset with dynamic 3D objects, resulting in the ObjaverseDy dataset.

We perform extensive comparisons of both novel view video synthesis and 4D generation results with respective state-of-the-art methods on datasets with synthetic (ObjaverseDy, Consistent4D (Jiang et al., 2024c)), and real-world (DAVIS (Perazzi et al., 2016; Pont-Tuset et al., 2017; Caelles et al., 2019)) data. We modify the FVD metric (Unterthiner et al., 2018) to evaluate both video frame and view consistency, validating the effectiveness and design choices of our approach. Fig. 1 shows some sample results of our approach.

To summarize, our contributions include:

- A novel SV4D network that can simultaneously reason across both frame and view axes. To our knowledge, this is the first work that trains a single novel view video synthesis network using 4D datasets, that can jointly perform novel view synthesis as well as video generation.
- A mixed sampling scheme that enables sequential processing of arbitrary long input videos while also retaining the multi-frame and multi-view consistency.
- State-of-the-art results on multiple benchmark datasets with both novel view video synthesis as well as 4D generation.

## 2 RELATED WORK

**3D Generation**. Here, we refer to the works that generate static 3D content as 3D generation. DreamFusion (Poole et al., 2023) first proposed to distill priors from the 2D diffusion model via SDS loss to optimize the 3D content from text or image. Several subsequent works (Yi et al., 2024; Tang et al., 2024; Shi et al., 2024a; Wang et al., 2024d; Li et al., 2024c; Weng et al., 2023; Pan et al., 2024a; Chen et al., 2024b; Sun et al., 2024a; Sargent et al., 2024; Liang et al., 2024b; Zhou et al., 2024; Guo et al., 2023) try to solve issues caused by the original SDS loss, such as multi-face Janus, slow generation speed, and over-saturated/smoothed generations. Recent works (Hong et al., 2024; Jiang et al., 2024a; Wang et al., 2024b; Zou et al., 2024; Wei et al., 2024; Tochilkin et al., 2024) try to directly predict the 3D model of an object via a large reconstruction model. Another approach (Liu et al., 2023b; 2024b; Long et al., 2024; Voleti et al., 2024; Ye et al., 2024; Karnewar et al., 2023; Li et al., 2024b; Shi et al., 2024b; 2023; Wang & Shi, 2023; Liu et al., 2023a; 2024a) to 3D generation is generating dense multi-view images with sufficient 3D consistency. 3D content is reconstructed based on the dense multi-view images. We follow this strategy, but generate consistent multi-view videos (instead of images) and then reconstruct the 4D object.

**Video Generation**. Recent video generation models (Ho et al., 2022; Voleti et al., 2022; Blattmann et al., 2023b;a; He et al., 2022; Singer et al., 2023a; Guo et al., 2024) have shown very impressive performance with consistent geometry and realistic motions. Video generation models have good generalization capabilities, as they are trained on large-scale image and video data that are easier to collect than large-scale 3D or 4D data. Hence, they are commonly used as foundation models for various generation tasks. Well-trained video generative models have shown their potential to generate multi-view images as a 3D generator (Voleti et al., 2024; Chen et al., 2024c; Han et al., 2024; Kwak et al., 2024; Melas-Kyriazi et al., 2024). SV3D (Voleti et al., 2024) adapts SVD (Blattmann et al., 2023a) to generate novel multiple views. In this work, we leverage the pre-trained video generation model for 4D generation by adding an additional view attention layer to align the multiview images.

**4D Generation**. On one hand, recent optimization-based methods (Singer et al., 2023b; Bahmani et al., 2024; Ling et al., 2024; Ren et al., 2023; Zhao et al., 2023; Jiang et al., 2024c; Zeng et al., 2024; Yin et al., 2023; Chen et al., 2024a; Rahamim et al., 2024; Geng et al., 2024; Yu et al., 2024) can generate 4D content by distilling pre-trained diffusion models in a 4D representation (Cao & Johnson, 2023; Kerbl et al., 2023; Mildenhall et al., 2021) via SDS loss Poole et al. (2023). However, they tend to take hours to generate 4D content due to time-consuming optimization. On the other hand, photogrammetry-based methods (Yang et al., 2025; Pan et al., 2024b) mimic a 3D object capture pipeline by directly generating multi-frame multi-view images of a 4D content with dynamic and multi-view consistency and then directly reconstructing 4D representations with them. Although these methods have much faster speeds, inference-only pipelines are adopted due to the paucity of the 4D data, thus making the spatial-temporal consistency still unsatisfactory. In this work, we proposed a monolithic model to generate a more consistent image grid, and we used a carefully curated 4D dataset to train the model. Recently, there has been a surge of interest in 4D generation, resulting in the development of concurrent works (Liang et al., 2024a; Zhang et al., 2024; Li et al., 2024a; Ren et al., 2024; Wang et al., 2024c; Jiang et al., 2024b; Sun et al., 2024b; Zhao et al., 2025; Wang et al., 2024a; Wu et al., 2024b; Zhu et al., 2025; Bai et al., 2024; Sun et al., 2025). Diffusion4D (Liang et al., 2024a) only generates diagonal images (space-time), by fine-tuning a 4D-aware video diffusion model with motion magnitude guidance. Vivid-ZOO (Li et al., 2024a) combines the 3D and video generative model and then finetunes alignment modules to mitigate the incompatibility between reused layers. 4Diffusion (Zhang et al., 2024) inserts temporal layers to a 3D-aware diffusion model to generate novel view videos. Both Vivid-ZOO and 4Diffusion freeze pre-trained weights, limiting their ability to fully capture spatial-temporal information. Additionally, they generate low-resolution images or a limited number of frames due to memory constraints. In contrast, SV4D finetunes all layers of our unified model, allowing for comprehensive spatial-temporal learning, and uses a mixed sampling scheme, enabling the generation of high-resolution and long novel views.

## 3 METHOD

Our main idea is to build multi-frame and multi-view consistency in a 4D object by surgically combining the frame-consistency in a video diffusion model, with the multi-view consistency in a multi-view diffusion model. In our case, we choose SVD (Blattmann et al., 2023a) and SV3D (Voleti et al., 2024) as the video and multi-view diffusion models respectively, for the advantages reasoned below. However, it is to be noted that any choice of attention-based diffusion models should work.

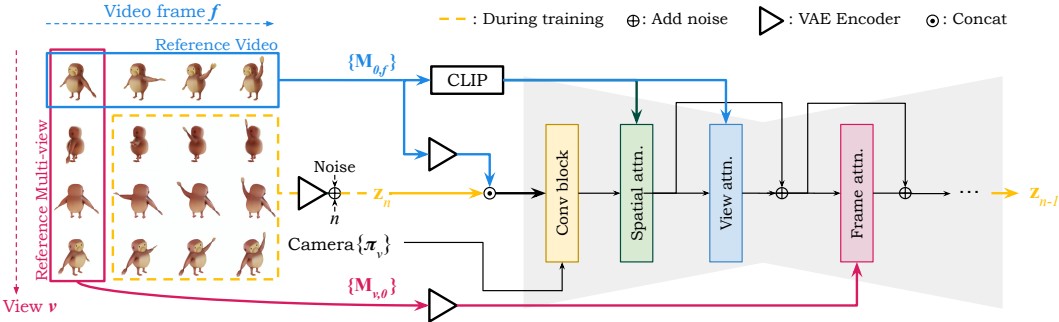

Figure 2: **SV4D model architecture.** For camera conditioning, we feed the sinusoidal embedding of camera viewpoints to the convolutional blocks in the UNet, and use the input video for cross-attention conditioning in the spatial and view attention blocks. To improve temporal consistency, we introduce an additional frame attention block, conditioned on the corresponding views of the first frame.

### 3.1 Novel View Video Synthesis via SV4D

**Problem Setting.** Formally, we begin with a monocular input video $\mathbf{J} \in \mathbb{R}^{F \times D}$ of a dynamic object of $F$ frames and $D := 3 \times H \times W$ dimensions, where $H$ and $W$ are the height and width of each frame. Our goal is to generate an image matrix $\mathbf{M} \in \mathbb{R}^{V \times F \times D}$ of the 4D object consisting of $V$ camera views and $F$ dynamic frames of each frame. Similar to SV3D (Voleti et al., 2024), the multi-view frames follow a camera pose trajectory $\boldsymbol{\pi} \in \mathbb{R}^{V \times 2} = \{(e_v, a_v)\}_{v=1}^{V}$ as a sequence of $V$ tuples of elevation $e$ and azimuth angles $a$. We generate this image matrix by iteratively denoising samples from a learned conditional distribution $p(\mathbf{M}|\mathbf{J}, \boldsymbol{\pi})$, parameterized by a 4D diffusion model. Due to memory limitation in generating $V \times F$ images simultaneously, we break down the full generation process into multiple submatrix generation steps.

**SV4D Network.** Our goal is to make the generated image matrix $\mathbf{M}$ dynamically consistent in the "**frame**" axis, and multi-view consistent in the "**view**" axis (see Fig. 2 left). To achieve this, we condition the image matrix generation on the corresponding **frames** of the monocular input video $\{\mathbf{M}_{0,f}\} = \mathbf{J}$ as well as the **views** from reference multi-view images of the first video frame $\{\mathbf{M}_{v,0}\}$, which provide motion and multi-view information of the 4D object, respectively. Without loss of generality, we obtain the reference multi-view images of the first input frame by sampling from a pre-trained SV3D (Voleti et al., 2024) model, which can be expressed as $p(\{\mathbf{M}_{v,0}\}|\{\mathbf{M}_{0,0}\}, \boldsymbol{\pi})$. The overall SV4D novel view video synthesis can be rewritten as sampling from the distribution $p(\mathbf{M}|\{\mathbf{M}_{0,f}\}, \{\mathbf{M}_{v,0}\}, \boldsymbol{\pi})$.

We build the SV4D network based on SVD (Blattmann et al., 2023a) and SV3D (Voleti et al., 2024) models to combine the advantages of both video and multi-view diffusion models. As illustrated in Fig. 2, SV4D consists of a UNet with multiple layers, where each layer contains a sequence of one residual block with Conv3D layers and three transformer blocks with attention layers: spatial, view, and frame attention. Similar to SV3D, the residual Conv block takes in the noisy latents of flattened image matrix as well as handles the incorporation of conditioning camera poses $\{\boldsymbol{\pi}_v\}$, and the spatial attention layer handles image-level details by performing attention across the image width and height axes. To better capture motion cues in the input monocular video $\{\mathbf{M}_{0,f}\}$, we concatenate its VAE latents to the noisy latents $\mathbf{z}_n$ before feeding it to the UNet.

For multi-view consistency, the view attention block transposes the features and performs attention in the multi-view axis. By using the CLIP embedding of corresponding input frames $\{\mathbf{M}_{0,f}\}$ as cross-attention conditioning, it allows the network to learn spatial consistency across novel views while maintaining the semantic context from the input video.

To further ensure dynamic consistency across video frames, we insert a frame attention layer in each UNet block, which applies the attention mechanism in the video frame dimension. The frame attention of each novel view video is conditioned on the corresponding reference view (of the first frame) $\{\mathbf{M}_{v,0}\}$ via cross-attention, allowing the network to preserve dynamic coherence starting from the first frame. We initialize the weights of the frame attention layers from SVD and the rest of

Figure 3: **SV4D model sampling** (7 frames and 3 views are for illustrative purposes only). To extend the generated multi-view videos while preserving temporal consistency, we propose a novel mixed-sampling strategy during inference. We first sample a sparse set of anchor frames, then use the anchor frames as new conditioning images to densely sample/interpolate the middle frames. To ensure a smooth transition between consecutive generations, we alternatively use the first (forward) or last (backward) frame within a time window for conditioning during dense sampling.

the network from $SV3D_p$, to leverage the generalizability as well as rich dynamic and multi-view priors learned from large-scale video and 3D datasets. More network details are in Appendix A.4.

**ObjaverseDy Dataset.** Considering that there exists no large-scale training datasets with dynamic 3D objects, we curate a new 4D dataset from the existing Objeverse dataset (Deitke et al., 2023b;a), a massive dataset with annotated 3D objects. Objaverse includes animated 3D objects, however, several of these animated 3D objects are not suitable for training due to having too few animated frames or insufficient motion. In addition, in the rendering stage, the common rendering and sampling setting may cause some issues. For example, dynamic objects may be out of the image if the camera distance is fixed because they have global motion; the motion of objects may be too fast or too slow if the temporal sampling step is fixed.

We follow several steps to curate and clean the 4D objects for our training purposes. We first filter out the objects based on a review of licenses. Then, we remove the objects whose animated frames are too few. To further filter out objects with minimal motion, we subsample keyframes from each video and apply simple thresholding on the maximum *L1* distance between these frames as motion measurement. To render the training novel view videos, we flexibly choose the camera distance from the object. Starting from a base value, we increase the camera distance until the object fits within all frames of the rendered images. We also dynamically adjust the temporal sampling step. Starting from a base value, we increase the sampling step until the *L1* distance between consecutive keyframes exceeds a certain threshold. These steps ensure a high-quality collection of 4D objects, with rendered multi-view videos that form our ObjaverseDy dataset. More dataset details are in Appendix A.1.

**Training Details.** We train SV4D on our ObjaverseDy dataset. We choose to finetune from the $SV3D_p$ (Voleti et al., 2024) model to output 40 frames ($F = 5$ frames along each of the $V = 8$ views) with the spatial resolution $576 \times 576$, where the parameters in the frame attention layers are initialized from SVD-xt (Blattmann et al., 2023a). Similar to SV3D, we train SV4D progressively by first training on the static camera orbits for 40K iterations, then fine-tuning it for 20K iterations on the dynamic orbits. We use an effective batch size of 16 during training on 2 nodes of 8 80GB H100 GPUs. For more training details, please see Appendix A.2.

**Inference Sampling Scheme.** Due to memory limitations, we cannot generate all the novel view frames at once. To generate the full $V \times F$ image matrix with arbitrary length videos, one can naively run the submatrix generation independently. However, we observe that it often leads to severe artifacts due to the inconsistencies between consecutive submatrices. Hence, we design a novel sampling scheme to mitigate the issue. The proposed sampling scheme is illustrated in Fig. 3. Note that we only show motion frame extensions here for illustrative purposes. We first generate a sparse set of anchor frames with SV4D (*interleaved sampling*) as shown in Fig. 3 (left). Then we use the anchor frames as new reference views to densely sample the remaining frames (*dense sampling*). To ensure a smooth transition between consecutive generations, we alternatively use the first (forward) or last (backward) anchor frame for conditioning at each diffusion step, as shown in Fig. 3 (middle). In experiments (Fig. 8), we demonstrate that our sampling scheme can generate novel view videos that are more temporally consistent compared to using an off-the-shelf video interpolation model (Li et al., 2023) to interpolate in between the SV4D-generated anchor frames.

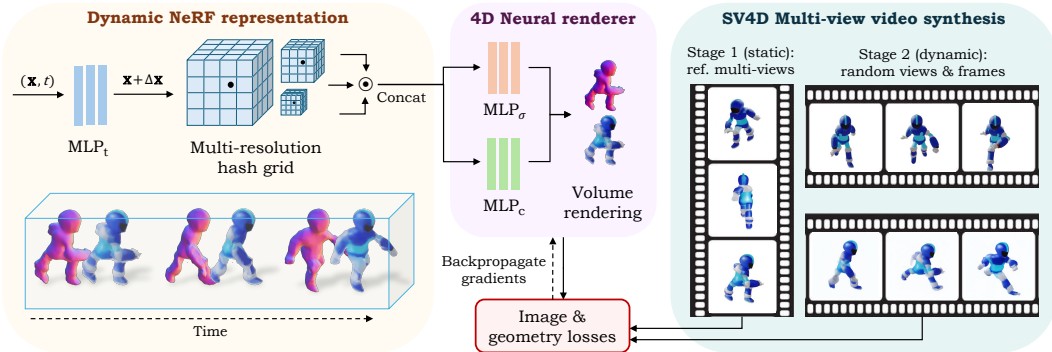

Figure 4: **Overview of optimization framework.** We first use the reference multi-view images (of the first frame) to optimize a static NeRF represented by a multi-resolution hash grid as well as density and color MLPs. Then, we unfreeze the temporal deformation MLP and optimize the dynamic NeRF with randomly sampled views and frames.

## 3.2 4D OPTIMIZATION FROM SV4D GENERATED NOVEL VIEW VIDEOS

**4D Representation.** As shown in Fig. 4, given the novel view videos generated by SV4D, we optimize a 4D representation to reconstruct the dynamic 3D asset. Formally, we learn a neural representation $\Psi_\theta : (\mathbf{x}, t) \mapsto (\sigma, c)$ that maps the sampled 3D points $\mathbf{x} = (x, y, z)$ along a camera ray, and the time embedding $t$ (i.e. continuous version of "frame" in the earlier discrete case) to its volumetric density $\sigma \in \mathbb{R}_+$ and color $c \in \mathbb{R}^3_+$. Similar to D-NeRF (Pumarola et al., 2021), we represent a 4D object by the composition of a canonical NeRF that captures the static 3D appearance and a deformation field which handles the object motion across time. For each 3D point $\mathbf{x}$ in the canonical space, we trilinearly interpolate the multi-resolution hash grid features following Instant-NGP (Müller et al., 2022), and decode them as density and color via $\text{MLP}_\sigma$ and $\text{MLP}_c$, respectively. The deformation field is represented by an MLP network $\text{MLP}_t$ conditioned on time embedding $t$, mapping temporal deformation of $\mathbf{x}$ to the common canonical space $\mathbf{x} + \Delta\mathbf{x}$. Overall, the dynamic NeRF parameters $\theta$ include the canonical hash grid, $\text{MLP}_t$, $\text{MLP}_\sigma$, and $\text{MLP}_c$. Since we do not exhaustively sample temporal timestamps or dense spatial views like in prior SDS-based approaches, we observe that this dynamic NeRF representation produces better 4D results compared to other representations such as 4D Gaussian Splatting (Wu et al., 2024a), which suffers from flickering artifacts and does not interpolate well across time or views. We show several examples to demonstrate this in the supplementary rebuttal video.

**Optimization Details.** By leveraging the consistent image matrix generated by SV4D as pseudo ground-truths, we adopt a simple photometry-based optimization without the cumbersome SDS losses. The reconstruction losses include a pixel-level MSE loss, mask loss, and a perceptual LPIPS (Zhang et al., 2018b) loss. Similar to SV3D (Voleti et al., 2024), we also use several geometric priors to regularize the output shapes, such as a mono normal loss similar to MonoSDF (Yu et al., 2022) as well as bilateral depth and normal smoothness losses (Voleti et al., 2024) to encourage smooth 3D surfaces where the projected image gradients are low.

For training efficiency and stability, we follow a coarse-to-fine, static-to-dynamic strategy to optimize a 4D representation. That is, we first freeze the deformation field $\text{MLP}_t$ and only optimize the canonical NeRF on the multi-view images of the first frame, while gradually increasing the rendering resolution from 128×128 to 512×512. Then, we unfreeze $\text{MLP}_t$ and randomly sample 4 frames × 4 views for training. Following the static-to-dynamic strategy, we also gradually optimize the time embedding $t$ from low to high temporal frequency. In our experiments, we find that sampling more timestamps in one batch and progressive optimization techniques are crucial to 4D output quality. We render the dynamic NeRF at 512×512 resolution and use an Adam (Kingma & Ba, 2014) optimizer to train all model parameters. The overall optimization takes roughly 15-20 minutes per object. More implementation details can be found in Appendix A.5.

Table 1: **Evaluation of novel view video synthesis on the Consistent4D dataset.** SV4D can achieve better video frame consistency while maintaining comparable image quality. [†] Our reproduced version.

| Model | LPIPS↓ | CLIP-S↑ | FVD-F↓ |
|---|---|---|---|
| SV3D (Voleti et al., 2024) | **0.129** | 0.925 | 989.53 |
| 4Diffusion (Zhang et al., 2024) | 0.164 | 0.863 | - |
| Diffusion[2] (Yang et al., 2025) | 0.189 | 0.907 | 1205.16 |
| STAG4D[†] (Zeng et al., 2024) | 0.131 | **0.929** | 861.88 |
| SV4D | **0.129** | **0.929** | **677.68** |

Table 2: **Evaluation of 4D outputs on the Consistent4D dataset.** SV4D can achieve better visual quality and video frame smoothness.

| Model | LPIPS↓ | CLIP-S↑ | FVD-F↓ |
|---|---|---|---|
| Consistent4D (Jiang et al., 2024c) | 0.160 | 0.87 | 1133.93 |
| STAG4D (Zeng et al., 2024) | 0.126 | 0.91 | 992.21 |
| 4Diffusion (Zhang et al., 2024) | 0.165 | 0.88 | - |
| Efficient4D (Pan et al., 2024b) | 0.130 | **0.92** | - |
| 4DGen (Yin et al., 2023) | 0.140 | 0.89 | - |
| DG4D (Ren et al., 2023) | 0.160 | 0.87 | - |
| GaussianFlow (Gao et al., 2024) | 0.140 | 0.91 | - |
| SV4D | **0.118** | 0.92 | **732.40** |

Table 3: **Evaluation of novel view video synthesis on the ObjaverseDy dataset.** SV4D can achieve superior performance in both video frame and multi-view consistency. [†] Our reproduced version.

| Model | LPIPS↓ | CLIP-S↑ | FVD-F↓ | FVD-V↓ | FVD-Diag↓ | FV4D↓ |
|---|---|---|---|---|---|---|
| SV3D (Voleti et al., 2024) | **0.131** | **0.920** | 729.67 | 375.49 | 526.78 | 690.49 |
| Diffusion[2] (Yang et al., 2025) | 0.188 | 0.869 | 1048.47 | 564.80 | 938.98 | 1320.29 |
| STAG4D[†] (Zeng et al., 2024) | 0.133 | 0.917 | 652.43 | 469.07 | 636.83 | 546.56 |
| SV4D | 0.136 | **0.920** | **585.09** | **331.94** | **503.02** | **470.46** |

# 4 EXPERIMENTS

**Datasets.** We evaluate SV4D-synthesized novel view videos and 4D optimization results on the synthetic datasets ObjaverseDy and Consistent4D (Deitke et al., 2023b). Consistent4D dataset contains dynamic objects collected from Objaverse (Deitke et al., 2023b;a). We excluded these objects from our training set to make a fair comparison. We used the same input video and evaluated views as Consistent4D. For the visual comparison, we also used single-view videos from the real-world videos dataset DAVIS (Perazzi et al., 2016; Pont-Tuset et al., 2017; Caelles et al., 2019).

**Metrics.** We use the SV4D model to generate multiple novel view videos corresponding to the trajectories of the ground truth camera in the evaluation datasets. We compare each generated image with its corresponding ground-truth, in terms of Learned Perceptual Similarity (*LPIPS* (Zhang et al., 2018b)) and CLIP-score (*CLIP-S*) to evaluate visual quality. In addition, we evaluate the video coherence by reporting FVD (Unterthiner et al., 2018), a video-level metric commonly used in video generation tasks. We calculate FVD with different ways (see Fig. 7): *FVD-F*: calculate FVD over frames at each view. *FVD-V*: calculate FVD over views at each frame. *FVD-Diag*: calculate FVD over the diagonal images of the image matrix. *FV4D*: calculate FVD over all images by scanning them in a bidirectional raster order.

**Baselines.** For *novel view video synthesis*, we compare SV4D with several recent methods capable of generating multiple novel view videos from a single-view video, including SV3D (Voleti et al., 2024), Diffusion[2] (Yang et al., 2025), STAG4D (Zeng et al., 2024). It is to be noted that all of these methods are inference-only techniques, and do not involve direct training in the 4D space like our method. We run SV3D to generate multi-view images for each video frame separately. STAG4D used Zero123++ (Shi et al., 2023) as the multi-view generator, which fixed the view angle, and hence the novel views generated from STAG4D cannot be changed to be consistent with the views evaluated. We reproduced STAG4D with SV3D as the multi-view generator. SV3D has been shown to generate more consistent 3D results than Zero123++, so this serves as a stronger baseline. For reference, we also compare SV4D with the concurrent work 4Diffusion (Zhang et al., 2024). We do not report the *FVD-F* metric for 4Diffusion, as it only generates 8 frames, which is inconsistent with the number of frames used in our evaluation. Note that the *FVD-F* metric is highly sensitive to the number of frames. For *4D generation*, we compare SV4D with other methods that can generate 4D representations, including Consistent4D (Jiang et al., 2024c), STAG4D (Zeng et al., 2024), 4Diffusion (Zhang et al., 2024), DreamGaussian4D (DG4D) (Ren et al., 2023), GaussianFlow (Gao et al., 2024), 4DGen (Yin et al., 2023), Efficient4D (Pan et al., 2024b). More baseline details are in Appendix C.

## 4.1 QUANTITATIVE COMPARISON

**Novel View Video Synthesis.** We quantitatively compare our method with the baselines in terms of novel view video synthesis results. Table 1 reports the comparisons in the Consistent4D dataset.

Table 4: **Evaluation of 4D outputs on the ObjaverseDy dataset.** SV4D consistently outperforms baselines in terms of all metrics, demonstrating superior performance in visual quality (*LPIPS* and *CLIP-S*), video frame consistency (*FVD-F*), multi-view consistency (*FVD-V*), and multi-view video consistency (*FVD-Diag* and *FV4D*).

| Model | LPIPS↓ | CLIP-S↑ | FVD-F↓ | FVD-V↓ | FVD-Diag↓ | FV4D↓ |
|---|---|---|---|---|---|---|
| Consistent4D (Jiang et al., 2024c) | 0.165 | 0.896 | 880.54 | 488.38 | 741.52 | 871.95 |
| STAG4D (Zeng et al., 2024) | 0.158 | 0.860 | 929.10 | 453.62 | 663.50 | 1003.16 |
| DreamGaussian4D (Ren et al., 2023) | 0.152 | 0.897 | 697.80 | 450.58 | 615.68 | 638.15 |
| SV4D | **0.131** | **0.905** | **659.66** | **368.53** | **525.65** | **614.35** |

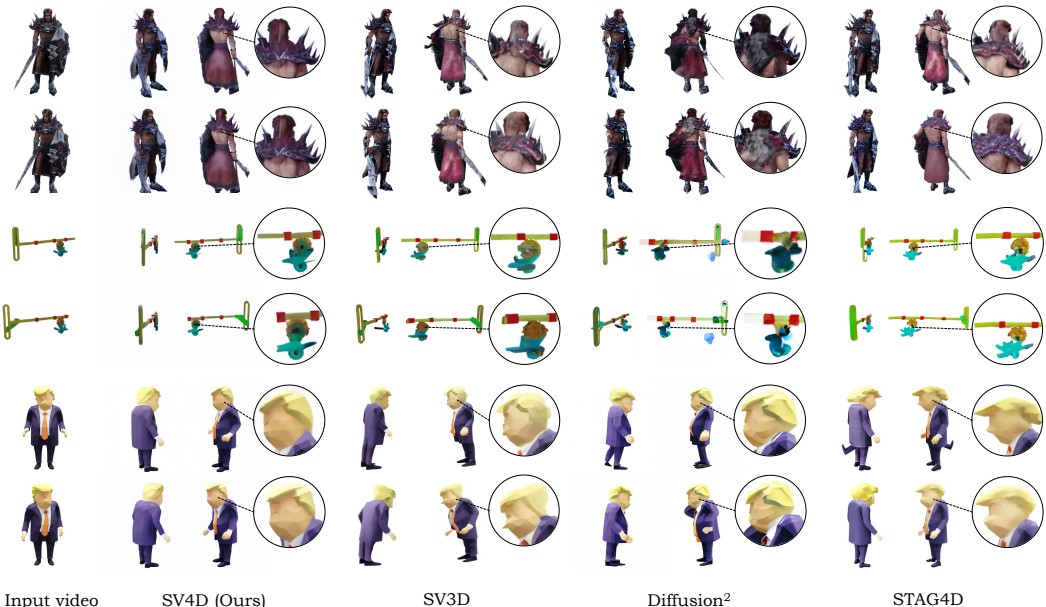

Input video     SV4D (Ours)     SV3D     Diffusion[2]     STAG4D

Figure 5: **Visual comparison of novel view video synthesis results.** We show two frames in the input videos and two novel-view results of the corresponding frames. Compared to the baseline methods, SV4D outputs contain geometry and texture details that are more faithful to the input video and consistent across frames.

Due to the fact that the evaluated views are too sparse, we only report *FVD-F* in the Consistent4D dataset. Our method can achieve much better performance in terms of video frame consistency while maintaining comparable performance in terms of image quality. In particular, our approach has a significant reduction of 31.5% and 21.4% in *FVD-F* compared to SV3D and STAG4D, respectively. Table 3 reports the comparisons in the Objaverse dataset. Our method can achieve much better video frame consistency as well as multi-view consistency while maintaining comparable performance in terms of image quality. In particular, our approach has much lower *FVD-F* compared to baseline methods, demonstrating our generated videos are much smoother. In addition, SV4D achieves better *FVD-V* which shows better multi-view consistency. Our method also surpasses the previous state-of-the-art methods in terms of *FVD-Diag* and *FV4D*, proving that the synthesized novel view videos have better video frame and multi-view consistency.

**4D Generation.** We quantitatively compare our optimized 4D outputs with the baselines in Consistent4D and ObjaverseDy dataset, as shown in Tables 2 and 4, respectively. Our method consistently outperforms baselines in terms of all metrics, demonstrating superior performance in visual quality (*LPIPS* and *CLIP-S*), motion smoothness (*FVD-F*), multi-view smoothness (*FVD-V*), and motion-multi-view joint smoothness (*FVD-Diag* and *FV4D*).

## 4.2 VISUAL COMPARISON

**Novel View Video Synthesis.** In Fig. 5 we show the visual comparison of our multi-view video synthesis results against other methods. We observe that applying SV3D frame-by-frame leads to

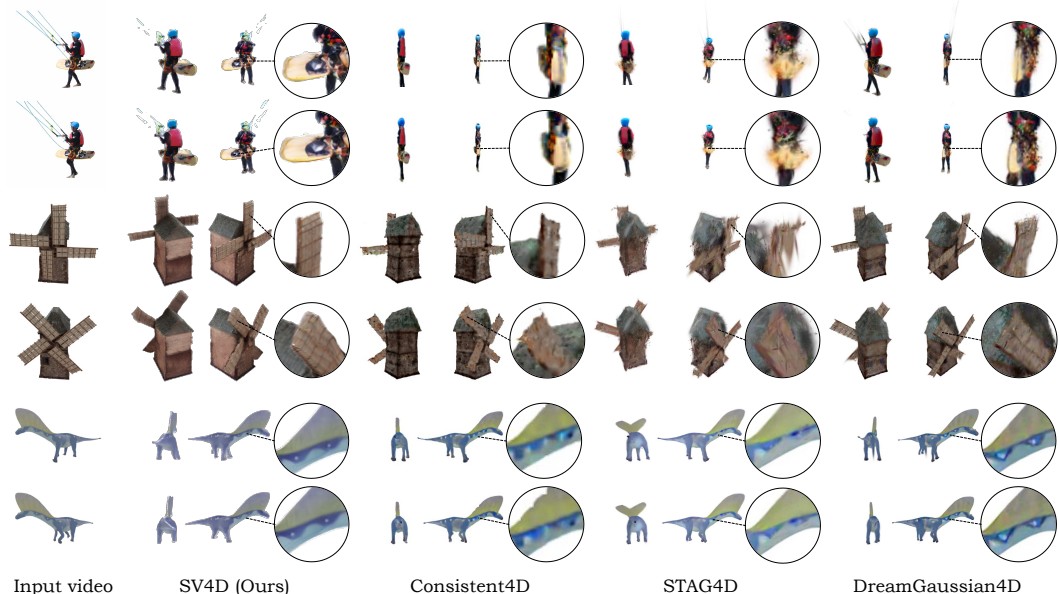

Figure 6: **Visual comparison of generated 4D outputs.** We show two frames in the input videos and render two novel views of the corresponding frames. By leveraging the consistent multi-view videos generated by SV4D, we can learn higher-quality 4D assets compared to the prior works with SDS-based losses. Our results are more detailed, consistent, and faithful to the input videos.

inconsistent geometry and texture in novel view videos. Diffusion$^2$ slightly improves the temporal coherence but tends to produce blurry or flickering artifacts. STAG4D can produce smoother videos but often fails at capturing large motion. Compared to these methods, SV4D can generate high-quality multi-view videos that are detailed, faithful to input videos, and temporally consistent.

**4D Generation.** We compare our generated 4D results with the prior methods in Fig. 6. For each video, we render the 4D outputs at two different timestamps and two novel views. Consistent4D and STAG4D often produce blurry outputs with inconsistent geometry and texture. DreamGaussian4D can generate finer texture details but still suffers from flickering artifacts and sometimes creates inaccurate geometry. Moreover, all these methods rely on the computationally expensive SDS loss, which is prone to spatially incoherent results and over-saturated texture. On the contrary, SV4D optimizes 4D assets using purely photometric and geometric losses, resulting in smoother videos with realistic and faithful object appearance. More visual comparison results can be found in Appendix B.1.

### 4.3 USER STUDY

In addition to the quantitative evaluation, we also conduct two user studies on our multi-view video synthesis results and 4D outputs, respectively. Concretely, we randomly select 10 real-world videos from the DAVIS dataset and 10 synthetic videos from the Objaverse or Consistent4D datasets. For each video, we randomly choose a novel camera view and ask the user to compare the novel view videos generated by 4 different methods (SV4D and 3 baselines). The users are asked to select a video that "looks more stable, realistic, and closely resembles the reference subject". For multi-view video synthesis, SV4D results are preferred 73.3% over per-frame SV3D (13.7%), Diffusion$^2$ (5.0%), and STAG4D (8.0%) among multiple participants. For the optimized 4D outputs, SV4D achieves 60% overall preference against Consistent4D (12.7%), STAG4D (9.7%), and DreamGaussian4D (17.6%) among multiple users.

### 4.4 ABLATIVE ANALYSES

We conduct several ablation experiments to validate the effectiveness of our model's design choices. We summarize the key findings below.

**SV4D can generate anchor frames with better consistency.** To validate the quality of anchor frames, we compare the anchor frames (8 views × 5 video frames) generated from SV4D and the

Table 5: **Evaluation of novel view video synthesis (anchor frames only)**. SV4D can effectively sample frames with faithful consistency and visual details. † Our reproduced version.

| | ObjaverseDy | | Consistent4D | |
|---|---|---|---|---|
| Model | FVD-F↓ | FV4D↓ | FVD-F↓ | FV4D↓ |
| SV3D (Voleti et al., 2024) | 700.72 | 831.18 | 656.59 | 780.99 |
| Diffusion² (Yang et al., 2025) | 896.98 | 890.70 | 801.99 | 1093.24 |
| STAG4D† (Zeng et al., 2024) | 765.79 | 669.28 | 601.23 | 659.79 |
| SV4D | **629.22** | **569.08** | **469.73** | **621.00** |

Table 6: **Evaluation of different sampling strategies on ObjaverseDy dataset.** SV4D sampling can effectively generate full image matrixes with faithful consistency and visual details.

| Sampling | LPIPS↓ | FVD-F↓ | FVD-Diag↓ | FV4D↓ |
|---|---|---|---|---|
| Interleaved | **0.136** | 731.28 | 567.15 | 717.21 |
| Independent | **0.136** | 663.41 | 512.14 | 488.31 |
| AMT (Li et al., 2023) | 0.140 | 612.61 | 505.36 | 472.87 |
| SV4D | **0.136** | **585.09** | **503.02** | **470.46** |

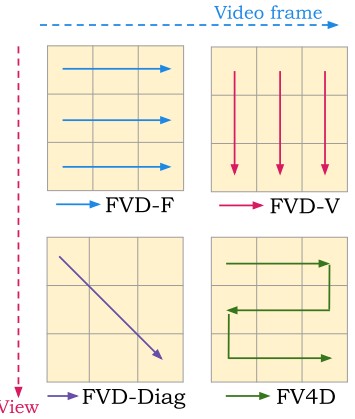

Figure 7: **Illustrations of video and 4D metrics.** FVD-F evaluates coherence between video frames from a fixed view. FVD-V captures multi-view consistency. We also design FVD-Diag and FV4D to evaluate 4D consistency by traversing the image matrix through different paths.

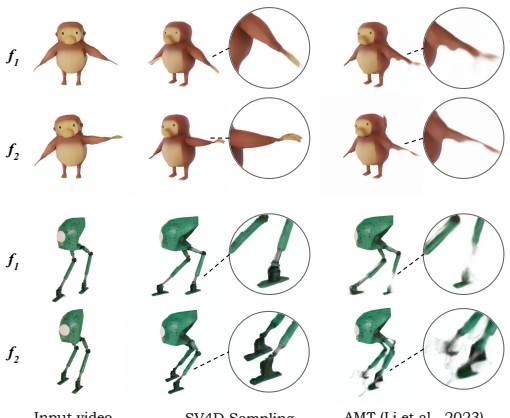

Figure 8: **Visual comparison of SV4D sampling and video interpolation.** Given the same sparse anchor frames, SV4D can effectively sample the middle frames with faithful motion and details, whereas an off-the-shelf video interpolation model creates blurry results or missing parts.

baselines on Consistent4D and ObjaverseDy dataset, as shown in Table 5. SV4D results have much lower *FVD-F* and *FV4D*, showing much better video frame and multi-view consistency.

**SV4D Sampling is better than off-the-shelf interpolation method.** Extending anchor frames to the full $V \times F$ image matrix is not trivial. In Table 6, we compare our SV4D sampling (*ours*) and the off-the-shelf interpolation method AMT (Li et al., 2023). The model with SV4D sampling outperforms AMT across all metrics. Fig. 8 validates this observation, where we can see that the SV4D sampling can synthesize distinct images while the results of the off-the-shelf interpolation method have a lot of blur results or missing parts. Table 6 also shows the results of two intuitive sampling strategies, independent and interleaved sampling, both of which produce videos with lower consistency. Appendix A.6 explains the details of independent and interleaved sampling. These results validate that the SV4D sampling is effective.

## 5 CONCLUSION

We present SV4D, a latent video diffusion model for novel view video synthesis and 4D generation. Given an input video, SV4D can generate multiple novel view videos that are dynamically and spatially consistent, by leveraging the dynamic prior in SVD and multi-view prior in SV3D within a unified architecture. The generated novel view videos can then be used to efficiently optimize a 4D asset without SDS losses from one or multiple diffusion models. Our extensive experiments show that SV4D outputs are more multi-frame and multi-view consistent than existing methods and generalizable to real-world videos, achieving the state-of-the-art performance on novel view video synthesis and 4D generation. We believe that SV4D provides a solid foundation model for further research on dynamic 3D object generation.

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

# A    IMPLEMENTATION DETAILS

## A.1    DATA DETAILS

Similar to SV3D (Voleti et al., 2024), we render views of a subset of the Objaverse dataset (Deitke et al., 2023b) curated CC-licensed animatable 3D objects from the original dataset. We held out some objects for evaluation purposes.

Each loaded object is scaled such the largest world-space XYZ extent of its bounding box is 1. We use Blender's CYCLES renderer to render multiple views and video frames for each object. We limit the number of samples in CYCLES renderer to save the rendering time. For lightning, we follow SV3D to randomly select from a set of curated HDRI envmaps.

For static orbits, we regularly sample azimuth angles with a constant value starting from azimuth 0. We randomly sample one elevation angle for all views. For dynamic orbits, the sequence of camera elevations for each orbit is obtained from a random weighted combination of sinusoids with different frequencies (Voleti et al., 2024). The azimuth angles are sampled regularly, and then a small amount of noise is added to make them irregular. The elevation values are smoothed using a simple convolution kernel and then clamped to a maximum elevation.

We then encode all of these images into latent space using SD2.1 (Rombach et al., 2022)'s VAE, and CLIP (Radford et al., 2021). We then store the latent and CLIP embeddings for all of these images along with the corresponding elevation and azimuth values, and frame index.

## A.2    SV4D TRAINING DETAILS

Our approach involves utilizing the widely used EMD (Karras et al., 2022), incorporating a $L2$ loss for fine-tuning, as followed in SVD (Blattmann et al., 2023b) and SV3D (Voleti et al., 2024). We adopt $L2$ loss during the training. To optimize the training efficiency and reduce GPU VRAM, we follow SV3D to reprocess and store the precomputed latent and CLIP embeddings for all images in advance. During training, these tensors are directly loaded rather than being computed in real-time. We found that with 8 views and 5 frames we were able to fit a batch size of 1 at $576 \times 576$ resolution. We randomly sample 8 views and 5 frames from our 21 rendered views and 21 frames for training.

## A.3    SV4D INFERENCE DETAILS

During inference, we use 50 steps of the deterministic DDIM sampler (Song et al., 2021). We adopt the SV4D sampling to generate the full V×F image matrix during the inference. We take $V = 8$ and $F = 21$ as an example. Our SV4D can generate 8 views × 5 frames in an inference. In the interleaved sampling, we first generate a sparse set of anchor frames with the frame indexes [4, 8, 12, 16, 20]. Then in the dense sampling, we use the anchor frames as new reference images to densely sample the remaining frames. For example, in the first run, we generate frames with the indexes [0, 1, 2, 3, 4] with frame 0 (forward) or frame 4 (backward) as conditioning. It takes about 40 seconds to generate 5 frames and 8 views for the SV4D model.

## A.4    SV4D NETWORK ARCHITECTURE

SV4D network consists of three attention layers: spatial attention, view attention, and frame attention. We reshape the feature $\mathbf{F}$ before feeding it to each attention layer $\gamma_{(\cdot)}(\cdot)$.

The spatial attention models spatial coherence across image spatial locations by calculating the attention of points at the same image:

$$\mathbf{F} = \text{rearrange}(\mathbf{F},\ B\ V\ F\ H\ W\ D \to (B\ V\ F)\ (H\ W)\ D) \tag{1}$$

$$\mathbf{F} = \gamma_s(\mathbf{F}) \tag{2}$$

$$\mathbf{F} = \text{rearrange}(\mathbf{F},\ (B\ V\ F)\ (H\ W)\ D \to\ B\ V\ F\ H\ W\ D) \tag{3}$$

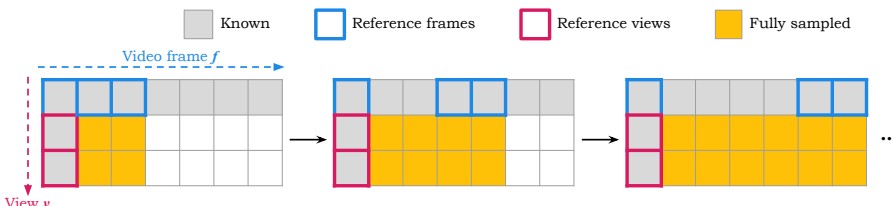

Figure 9: **Independent sampling**. The submatrix sampled at each inference consists of consecutive frames and each submatrix is sampled separately, without relying on previous fully sampled images.

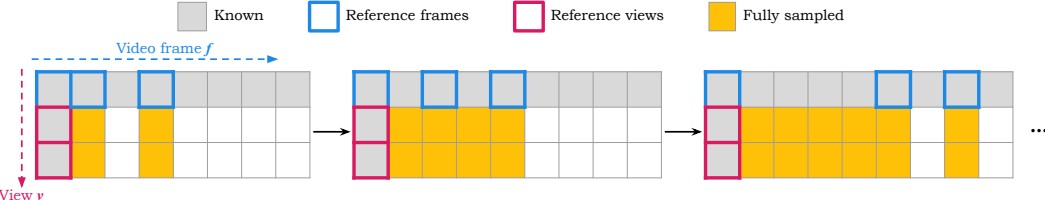

Figure 10: **Interleaved Sampling**. The submatrix sampled at each inference consists of interleaved frames and each submatrix is sampled separately, without relying on previous fully sampled images.

The view attention models coherence across views by calculating the attention of points at the same spatial location and the same frame across views:

$$\mathbf{F} = \text{rearrange}(\mathbf{F}, \; B \; V \; F \; H \; W \; D \to (B \; F \; H \; W) \; V \; D) \tag{4}$$

$$\mathbf{F} = \gamma_v(\mathbf{F}) \tag{5}$$

$$\mathbf{F} = \text{rearrange}(\mathbf{F}, \; (B \; F \; H \; W) \; V \; D \to \; B \; V \; F \; H \; W \; D) \tag{6}$$

The frame attention models coherence across frames by calculating the attention of points at the same spatial location and the same view across frames:

$$\mathbf{F} = \text{rearrange}(\mathbf{F}, \; B \; V \; F \; H \; W \; D \to (B \; V \; H \; W) \; F \; D) \tag{7}$$

$$\mathbf{F} = \gamma_f(\mathbf{F}) \tag{8}$$

$$\mathbf{F} = \text{rearrange}(\mathbf{F}, \; (B \; V \; H \; W) \; F \; D \to \; B \; V \; F \; H \; W \; D) \tag{9}$$

$B$ is batch size, $V$ is number of views, $F$ is number of frames. $H$, $W$, and $D$ are width, height, and feature dimension of the image feature. In our experiments, both $V$ and $F$ are 8 and 5, respectively.

Note that our frame attention layers are skip-connected to the outputs of view attention with a learnable blending weight per layer, allowing it to effectively merge the spatial and temporal information in the final output. Although an optimal way is to perform spatial-temporal attention jointly without considering memory limitation, we find that our sequential design can best leverage the priors in SVD and SV3D with minimal computation overhead.

## A.5 DYNAMIC NERF OPTIMIZATION DETAILS

Our main 4D reconstruction losses are the pixel-level mean squared error $\mathcal{L}_{\text{mse}} = \|\boldsymbol{M} - \hat{\boldsymbol{M}}\|^2$, LPIPS (Zhang et al., 2018a) loss $\mathcal{L}_{\text{lpips}}$, and mask loss $\mathcal{L}_{\text{mask}} = \|\boldsymbol{S} - \hat{\boldsymbol{S}}\|^2$, where $\boldsymbol{S}$, $\hat{\boldsymbol{S}}$ are the predicted and ground-truth masks. We further employ a normal loss using the estimated mono normal by Omnidata (Eftekhar et al., 2021), which is defined as the cosine similarity between the rendered normal $\boldsymbol{n}$ and estimated pseudo ground truths $\bar{\boldsymbol{n}}$: $\mathcal{L}_{\text{normal}} = 1 - (\boldsymbol{n} \cdot \bar{\boldsymbol{n}})$. To regularize the output geometry, we apply a smooth depth loss inspired by RegNeRF (Niemeyer et al., 2022): $\mathcal{L}_{\text{depth}}(i, j) = (d(i, j) - d(i, j + 1))^2 + (d(i, j) - (d(i + 1, j))^2$, where $i, j$ indicate the pixel coordinate. For surface normal we instead rely on a bilateral smoothness loss similar to Boss et al. (2022). We found that this is crucial to getting high-frequency details and avoiding over-smoothed surfaces. For this loss we compute the image gradients of the input image $\nabla \boldsymbol{I}$ with a Sobel filter (Kanopoulos et al., 1988). We then encourage the gradients of rendered normal $\nabla \boldsymbol{n}$ to be smooth if (and only if) the

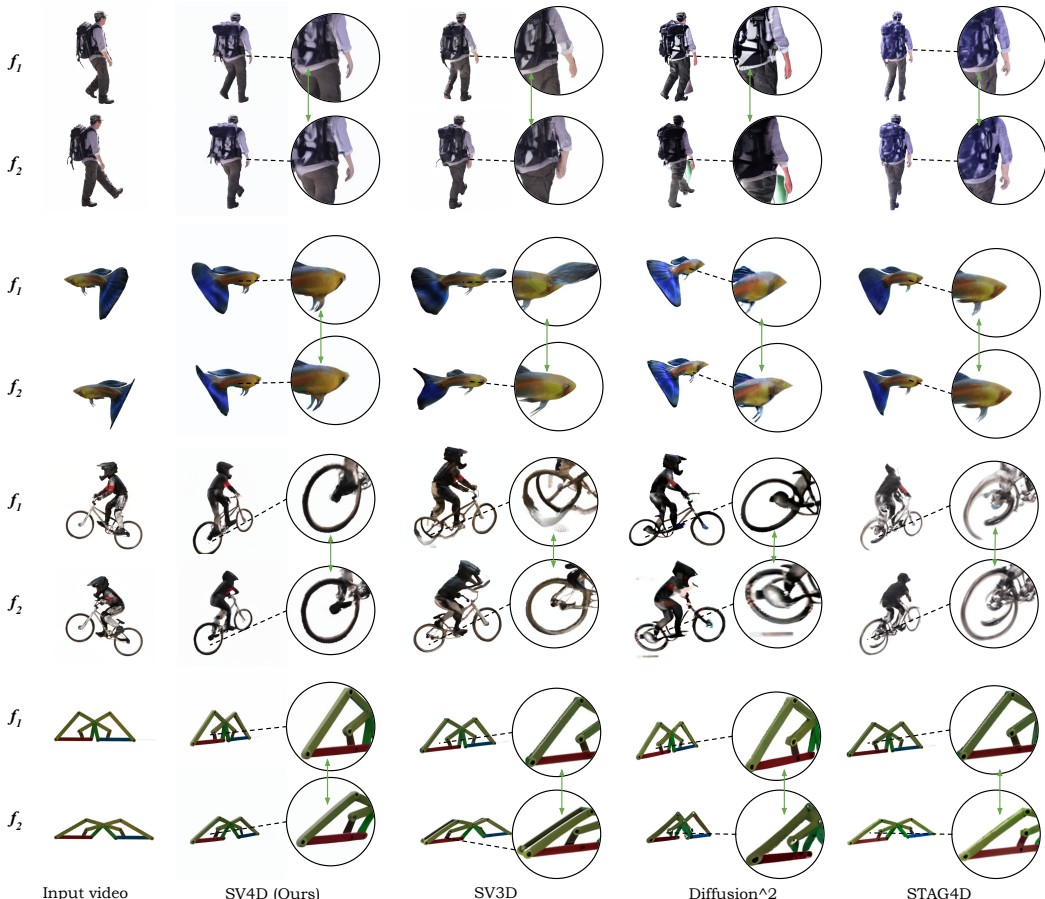

Figure 11: **Visual Comparison of Novel View Video Synthesis Results.** We show two frames in the input videos and one novel-view result of the corresponding frames. Compared to the baseline methods, SV4D outputs contain geometry and texture details that are more faithful to the input video and consistent across frames.

input image gradients $\nabla I$ are smooth. The loss can be written as $\mathcal{L}_{\text{bilateral}} = e^{-3\nabla I}\sqrt{1 + ||\nabla n||}$. The overall objective is then defined as the weighted sum of these losses. All losses are applied in both static and dynamic stages. We use an Adam optimizer (Kingma & Ba, 2014) with a learning rate of $0.01$ for both stages.

### A.6 DIFFERENT SAMPLING STRATEGIES

Due to memory limitations, we cannot generate all the novel view frames at once. To generate the full image matrix, one can run the submatrix generation in different strategies. To validate the effectiveness of our SV4D sampling, we compare ours with two other sampling strategies: independent sampling and interleaved sampling, which are shown in Fig. 9 and Fig. 10, respectively. For illustrative purposes only, the SV4D model can generate 2 frames and 2 views at each inference. In the independent sampling, the submatrix sampled at each inference consists of *consecutive* frames and each submatrix images are sampled separately, without relying on previous fully sampled images. In the interleaved sampling, the submatrix sampled at each inference consists of *interleaved* frames.

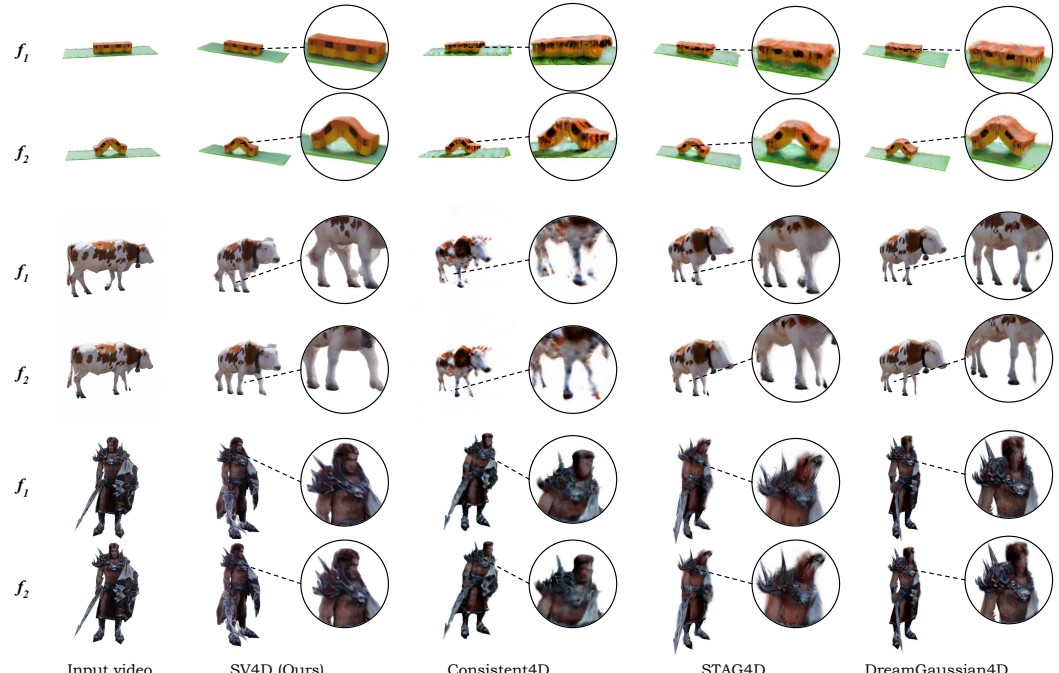

Figure 12: **Visual Comparison of Generated 4D Outputs.** We show two frames in the input videos and render one view result of the corresponding frames. By leveraging the consistent multi-view videos generated by SV4D, we can learn higher-quality 4D assets compared to the prior works with SDS-based losses. Our results are more detailed, consistent, and faithful to the input videos.

Table 7: **Additional Ablations.** *SV4D w/o frame atten.*: SV4D without frame attention. *SV4D w/o MV*: SV4D without reference multi-views. *SV4D with CLIP MV*: SV4D with CLIP embeddings (instead of VAE latents) of the first frame for frame attention conditioning and without reference multi-views.

| Model | LPIPS↓ | CLIP-S↑ | FVD-F↓ | FVD-V↓ | FVD-Diag↓ | FV4D↓ |
|---|---|---|---|---|---|---|
| SV4D w/o frame atten. | **0.131** | **0.920** | 729.67 | 375.49 | 526.78 | 690.49 |
| SV4D w/o MV | 0.135 | 0.870 | 876.07 | 410.90 | 608.93 | 710.66 |
| SV4D with CLIP MV | 0.142 | 0.868 | 1174.47 | 530.74 | 819.22 | 1327.75 |
| SV4D | 0.136 | **0.920** | **585.09** | **331.94** | **503.02** | **470.46** |

# B ADDITIONAL RESULTS

## B.1 MORE VISUAL COMPARISONS

We show more visual comparisons of novel view synthesis and 4D generation, as shown in Fig. 11 and Fig. 12. These results further demonstrate that SV4D-generated images and 4D results are more consistent and detailed, faithful to the conditioning videos compared to the prior works.

## B.2 MORE ABLATIVE RESULTS

We show an additional ablation study on the SV4D network architecture to justify our model design in Table 7. In particular, we report the quantitative comparison of ablated models: SV4D without frame attention, SV4D without multi-view conditioning, and SV4D with CLIP embeddings (instead of VAE latents) of the first frame for frame attention conditioning while removing the reference multi-views. The results justify that using VAE latents of reference multi-views as frame attention conditioning achieves better performance in most metrics compared to other design choices.

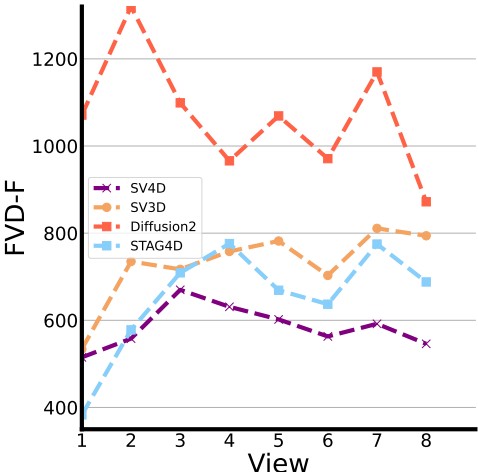

Figure 13: **FVD-F vs. View Number** on the ObjaverseDy dataset. We observe that SV4D produces better temporal consistency in almost all views.

### B.3 QUALITY PER VIEW

We plot the FVD-F value for each view on the ObjaverseDy dataset, as shown in Fig. 13. We observe that SV4D produces better temporal consistency in almost all views.

## C BASELINE DETAILS

For *novel view video synthesis* (Table 1, 3, and 5), we compare SV4D with SV3D (Voleti et al., 2024), Diffusion[2] (Yang et al., 2025), STAG4D (Yang et al., 2025), 4Diffusion (Zhang et al., 2024). The results of SV3D, Diffusion[2], and 4Diffusion are generated with their official code. STAG4D used Zero123++ (Shi et al., 2023) as the multi-view generator, which fixed the view angle, and hence the novel views generated from STAG4D cannot be changed to be consistent with the views evaluated. We reproduced STAG4D with SV3D as the multi-view generator. SV3D has been shown to generate more consistent 3D results than Zero123++, so this serves as a stronger baseline.

For *4D generation*, we compare SV4D with Consistent4D (Jiang et al., 2024c), STAG4D (Zeng et al., 2024), 4Diffusion (Zhang et al., 2024), DreamGaussian4D (DG4D) (Ren et al., 2023), GaussianFlow (Gao et al., 2024), 4DGen (Yin et al., 2023), Efficient4D (Pan et al., 2024b). On the Consistent4D dataset (Table 2), the results of Consistent4D, STAG4D, Efficient4D, and GaussianFlow are from the original paper. 4DGen results are from Efficient4D paper, and DG4D results are from GaussianFlow paper. The results of 4Diffusion are generated with its official code. On the ObjaverseDy dataset (Table 4), all results of our baselines are generated from their official code.

## D SUPPLEMENTARY VIDEO

Beyond the paper, our supplementary materials offer a comprehensive video that provides an in-depth introduction to our task and method. Additionally, we add more visual comparisons which further showcase the effectiveness of our approach.

