# OpenReview forum: "SV4D: Dynamic 3D Content Generation with Multi-Frame and Multi-View Consistency"
_ICLR.cc/2025/Conference — ICLR 2025 Poster_

### Official Review · Reviewer_wzad · 2024-10-31

**Soundness:** 3
**Presentation:** 3
**Contribution:** 3
**Rating:** 6
**Confidence:** 4

**Summary:**

This paper introduces Stable Video 4D (SV4D), a novel latent video diffusion model designed to generate dynamic, multi-view consistent 3D videos from a single input video. Unlike previous models that separately handled video generation and novel view synthesis, SV4D uses a unified approach to generate multi-frame, multi-view videos that retain both temporal and spatial consistency.  Extensive evaluations on synthetic and real-world datasets show SV4D’s superior performance in novel-view video synthesis and 4D generation.

**Strengths:**

1. SV4D combines video generation and novel view synthesis within a single framework, overcoming the limitations of separate models and ensuring multi-frame and multi-view consistency.
2. The use of view attention and frame attention blocks ensures alignment across different frames and viewpoints, enhancing dynamic consistency.

**Weaknesses:**

Although SV4D demonstrates promising results on selected datasets, its performance on a broader range of objects, particularly those with complex or intricate details, is not thoroughly evaluated.

**Questions:**

How does SV4D perform on complex, out of distribution high-detail objects or scenes with intricate textures For example, some of the scenes shown in L4GM website https://research.nvidia.com/labs/toronto-ai/l4gm/.

---

> ### Author Response · Authors · 2024-11-21
>
> We thank the reviewer for the valuable feedback. According to the suggestions, we have **updated our paper PDF** and **included a rebuttal video** in the Supplementary Material. We also address the additional comments and questions below.
>
> **W1: Objects with complex or intricate details**
> * We show additional results of SV4D with complex and intricate details in our rebuttal video (02:50 - 03:15).
> * Note that SV4D captures better geometry and texture details compared to prior published works (Consistent4D, STAG4D, DreamGaussian4D) with publicly available code.
>
> **Q1: Results on videos from L4GM website**
> * Please note that L4GM is a concurrent work and has not released their model/code.
> * Despite this, we show several results of SV4D on the L4GM demo videos in our rebuttal supplemental video (01:28 - 02:03), demonstrating comparable performance on detailed objects.
>
> In light of our point-by-point responses, it would be great if the reviewer could increase the score of our paper and champion it for publication. We thank the reviewer for the feedback and continued support for a quality publication.

---

> > ### Author Response · Authors · 2024-11-25
> >
> > We thank the reviewer again for the feedback on our paper!
> > As the discussion deadline (Nov. 26) is approaching, we were wondering whether the reviewer had the chance to look at our response and whether there is anything else the reviewer would like us to clarify. We sincerely hope that our response has addressed the concerns, and if so, we would be grateful if the reviewer could consider increasing the score accordingly.
> >
> > Best,
> > SV4D authors

---

> > > ### Author Response · Authors · 2024-11-27
> > >
> > > We thank Reviewer `wzad` once again for taking the time to review our work. We wanted to kindly check if Reviewer `wzad` had an opportunity to review our response and if there are any additional clarifications needed. We sincerely hope our response has addressed the concerns and would greatly appreciate it if Reviewer `wzad` could consider increasing the score accordingly.
> > >
> > > Best,
> > > SV4D authors

---

> ### Author Response · Authors · 2024-12-02
> **Awaiting Response from Reviewer wzad (Final Day Remaining)**
>
> We sincerely thank Reviewer `wzad` for taking the time to review our work. With just **one day remaining** until the deadline, we kindly request feedback on our response. If any part of our explanation is unclear, please let us know. We genuinely hope our response has addressed all concerns and would greatly appreciate it if the reviewer could consider raising the score accordingly.
>
> Best,
> SV4D authors

---

### Official Review · Reviewer_EfLA · 2024-11-01

**Soundness:** 3
**Presentation:** 3
**Contribution:** 3
**Rating:** 8
**Confidence:** 4

**Summary:**

This manuscript presents SV4D, a framework tailored for the generation of dynamic 3D content. The key contribution of this framework is a diffusion model characterized by a dual-attention design, capable of simultaneously sampling 5 frames $\times$ 8 views with satisfactory consistency in both dimensions. To generate videos of meaningful length within a reasonable memory budget, this work also proposes a mixed-sampling strategy, which includes interleaved sampling of anchor frames and dense sampling of middle frames conditioned on reference at both sides. For the training of this model, the authors meticulously curated a subset of Objaverse comprising high-quality animated 3D objects.

**Strengths:**

1.	The paper is overall well-organized and easy to follow.

2.	Their efforts and experience in data curation and training of the 4D generative model have sufficient value to the community, and it can be more valuable with more substantive ablation studies to validate the final design choices.

3.	The proposed mix-sampling strategy is interesting; it effectively extends the local spatial-temporal consistency to a broader scale while avoiding memory constraint.

**Weaknesses:**

1.	The main contribution of this work lies in the 4D diffusion model; however, there is no ablation on its design and training. It is important to demonstrate to which extent the results are non-trivial and could provide inspiration for subsequent research. In contrast, the ablation provided in this manuscript primarily focuses on the mixed-sampling strategy, which is , while necessary, more like a stopgap solutions to mitigate the memory limitation in this framework.

2.	It is encouraged to provide more generated 4D assets in supplementary videos.
Now the supplementary video is more like a introduction to the motivation and methodology of this work, which is slightly repetitive with the role of the main paper. The reviewer notices that there are only 12 generated dynamic objects shown in the video, each of them rendered in single fixed-view, which is insufficient to demonstrate multi-view consistency of generated assets.

**Questions:**

1.	The manuscript has not introduced the scale of ObjaverseDy. Does the author think that the scale of data is sufficient for training a diffusion model for object-centric 4D generation? Is there possibility or necessity for further scaling the data?

2.	L104 claims the capability to handle arbitrarily long input videos. Have the authors tested on substantively longer videos, or is this capability just theoretical?

3.	L264 says that "we only show motion frame extensions here for illustrative purposes." Does this imply that similar extensions could be applied to views?

---

> ### Author Response · Authors · 2024-11-21
>
> We thank the reviewer for the valuable feedback. According to the suggestions, we have **updated our paper PDF** and **included a rebuttal video** in the Supplementary Material. We also address the additional comments and questions below.
>
> **W1: Ablation study on SV4D network**
> * We perform ablation study in the table below to validate our design choices of SV4D network. The ablative results are also included in the Appendix B.2 and Table 7.
> * Specifically, we report the quantitative results of ablated models:
> SV4D without frame attention (*w/o frame atten.*),
> SV4D without multi-view conditioning (*SV4D w/o MV*), and
> SV4D with CLIP embeddings of the first frame for frame attention conditioning (*SV4D with CLIP MV*).
> * These results demonstrate that using VAE latents of reference multi-views for frame attention conditioning achieves the best image quality and spatial-temporal consistency.
>
> | Method                     | LPIPS↓ | CLIPS↑ | FVD-time↓ | FVD-view↓ | FVD-diagonal↓ | FV4D↓   |
> |----------------------------|--------|--------|-----------|-----------|---------------|---------|
> | SV4D                       | 0.136  | **0.920**  | **585.09**    | **331.94**    | **503.02**        | **470.46**  |
> | w/o frame atten.      | **0.131**  | **0.920**  | 729.67    | 375.49    | 526.78        | 690.49  |
> | SV4D w/o MV          | 0.135  | 0.870  | 876.07    | 410.90    | 608.93        | 710.66  |
> | SV4D with CLIP MV          | 0.142  | 0.868  | 1174.47   | 530.74    | 819.22        | 1327.75 |
>
> **W2: Additional 4D results**
> * We show more 4D results from multiple views in the rebuttal video provided in the supplementary material.
> * In the rebuttal video (00:00 - 00:52), we show the SV4D-generated 8 novel views on synthetic data and real-world data, demonstrating the multi-view consistency of SV4D outputs.
> * In the rebuttal video (01:28 - 02:03), we use the input videos from [L4GM](https://research.nvidia.com/labs/toronto-ai/l4gm/) website and show SV4D-generated 8 novel views, further demonstrating the spatial consistency from our method on detailed objects.
>
> **Q1: The scale of training dataset**
> * The Objaverse dataset contains 44k dynamic objects. We filter out nearly half of the objects based on licenses and amount of motion.
> * As mentioned in Sec 1: lines 88-92, although limited by the size of 4D training data, SV4D shows decent generalization capability since it inherits from SVD and SV3D the priors learned from large-scale video and multi-view data.
> * To further scale up the training dataset, joint training with video and 3D data poses an interesting future work.
>
> **Q2: Sampling long videos**
> * To generate long videos, we can run the proposed sampling in an autoregressive manner while keeping the same gap between anchor frames to maintain temporal coherence. For instance, we first sample anchor frames (0, 5, 10, 15, 20) to generate frames 0-20, then use the multi-views of frame 20 as the condition to generate new anchor frames (20, 25, 30, 35, 40).
> * We show example generations in the rebuttal video (00:52 - 01:07).
>
> **Q3: Sampling denser views**
> * Indeed, the proposed sampling strategy can be applied in the view axis to generate denser novel views.
> * We show an example result in the rebuttal video (01:08 - 01:28).
>
> In light of our point-by-point responses, it would be great if the reviewer could increase the score of our paper and champion it for publication. We thank the reviewer for the feedback and continued support for a quality publication.

---

> > ### Author Response · Authors · 2024-11-25
> >
> > We thank the reviewer again for the feedback on our paper!
> > As the discussion deadline (Nov. 26) is approaching, we were wondering whether the reviewer had the chance to look at our response and whether there is anything else the reviewer would like us to clarify. We sincerely hope that our response has addressed the concerns, and if so, we would be grateful if the reviewer could consider increasing the score accordingly.
> >
> > Best,
> > SV4D authors

---

> > > ### Comment · Reviewer_EfLA · 2024-11-26
> > >
> > > Thanks for the author’s efforts in addressing my concerns and questions. I am largely satisfied with the additional results provided during the rebuttal. I appreciate the contribution of this work as the first released 4D diffusion model, their exploration and experience could provide sufficient value for the community. My previous suggestions mainly aimed at encouraging more insights from the perspective of model training, which could be beneficial for future exploration along this path. The additional results can help achieve this goal. Therefore, I will keep my current positive recommendation and take active consideration for further raising my rating.

---

> > > > ### Author Response · Authors · 2024-11-26
> > > >
> > > > We thank Reviewer EfLA once again for the time and effort in reviewing our work and for providing such a positive and encouraging review. We are pleased that the reviewer’s concerns and questions have been addressed. The reviewer’s valuable suggestions have helped strengthen our work and offer deeper insights. We believe this work will **inspire further research into designing and training advanced 4D diffusion models**.
> > > >
> > > > Best,
> > > > SV4D authors

---

### Official Review · Reviewer_p3AT · 2024-11-02

**Soundness:** 3
**Presentation:** 2
**Contribution:** 3
**Rating:** 6
**Confidence:** 5

**Summary:**

This work makes attempts to build a multi-view video diffusion model conditioned on reference single-view video and reference multi-view images.  To inherit knowledge learned by previous 3D diffusion models and video diffusion models, the authors carefully design the multi-view video generation network to combine both of them. Specifically, they insert the frame-attention module of the video generation model to 3D generation model to achieve spatial and temporal consistency. To handle GPU memory problem when inferring long-videos, they propose a mixed-sampling scheme. Extensive experiments with regard to novel view video synthesis and 4D generation demonstrate the advantages of the proposed method.

**Strengths:**

1) The work makes attempts to train a mult-view video diffusion model for 4D generation, which is encouraging and could benifit the community.
 2) The authors test their model on both real-world and synthetic objects to prove the generalization ability of their model.
 3) The authors conduct extensive compairsons with previous mehtods.

**Weaknesses:**

1) network design: According to Figure 2, frame-attention is inserted after view-attention, and the output of frame-attention directly serves as the final result. However, since frame-attention only operates on time axis and is not aware of multi-view information, latents processed by it might suffer from multi-view inconsistency, and cannot directly serve as the final result. It is observed that the 4D reconstruction results suffer from blurry effect, and I suspect it is due to multi-view inconsistency of the generated videos.
2) problem setting: The proposed model takes generated reference video and generated multi-view images of the first frame as the input. However, it is possible that the two inputs have conflict with each other since they are generated independently. For example, if the video depicts a object is turning around, it may not be consisent with the generated multi-view images espicially in side views and back views. I don't see the authors show objects with such movements in the experiments.
3) experiments: I'm curious about the real-world video experiements. The authors only train their model on synthetic data, so the generailization to real-world cases might be difficult. I don't find many visualizations of real-world generated objects. Could the authors point out where the visualizations are?
4) Some citations regarding to concurrent works are not accurate, for example (Line245 Wang et al, 2024a) and (Line 149 4Diffusion Yang et al, 2024). The authors are suggested to further check them.
5) The authors claim they can handle arbitrary length videos; however, when the input video is very long, the anchor frames will have little temporal coherence and be out-of-distribution for the proposed model.

**Questions:**

1) training dataset: I can't find the size of the training dataset in the paper. (Maybe I miss it?) However, as far as I know, many animated objects in objaverse datasets are of low-quality, so I'm curious how many models is used in this work after data filtering.
2) I'm curious why the authors choose dynamic nerf as the 4D representations instead of dynamic gaussians.

---

> ### Author Response · Authors · 2024-11-21
>
> We thank the reviewer for the valuable feedback. According to the suggestions, we have **updated our paper PDF** and **included a rebuttal video** in the Supplementary Material. We also address the additional comments and questions below.
>
> **W1: Network design of frame attention**
> * Our frame attention layers are skip-connected to the outputs of multi-view attention with a learnable blending weight per layer, allowing it to effectively merge the spatial and temporal information in the final output. Although an optimal way is to perform spatial-temporal attention jointly without considering memory limitation, we find that our sequential design can best leverage the priors in SVD and SV3D with minimal computation overhead.
> * We added the clarification and discussion in the Appendix A.4.
>
> **W2: Reference video and reference multi-views**
> * The reviewer is absolutely right. Although independent generation of reference video and reference multi-views is commonly used in 4D generation methods (e.g. 4DGen, Diffusion^2, 4Real, L4GM, EG4D), we acknowledge the limitations of such an approach, especially in cases of heavy occlusion in the first frame or rotating motion in the reference video. In future work, we will continue exploring strategies to eliminate dependence on reference multi-views.
>
> **W3: Real-world videos**
> * The results on (segmented) real-world videos are shown in our rebuttal video (00:38 - 00:52, 02:29 - 02:39, 02:58 - 03:15). Since the SV4D model is built upon a pretrained SVD model, it inherits the generalization ability to real-world videos learned from large-scale video data in-the-wild.
>
> **W4: Inaccurate citations**
> * We thank the reviewer for pointing out the error, and we have corrected them in the revised draft.
>
> **W5: Long video generation**
> * To generate long videos, we can run the proposed sampling in an autoregressive manner while keeping the same gap between anchor frames to maintain temporal coherence. For instance, we first sample anchor frames (0, 5, 10, 15, 20) to generate frames 0-20, then use the multi-views of frame 20 as the condition to generate new anchor frames (20, 25, 30, 35, 40).
> * We show an example generation in the rebuttal video (00:52 - 01:07). Note that the object motion is often repetitive in most object-centric videos from existing datasets like Objaverse or Consistent4D, which preserves a strong temporal correspondence between sparse anchor frames.
>
> **Q1: Training dataset**
> * The Objaverse dataset contains 44k dynamic objects. We filter out nearly half of the objects based on licenses and amount of motion.
>
> **Q2: Dynamic NeRF v.s. Dynamic Gaussians**
> * As mentioned in lines 304-307, we observe that Dynamic Gaussian representation “suffers from flickering artifacts and does not interpolate well across time or views” compared to Dynamic NeRF.
> * We show additional results using a 4D Gaussian representation in the rebuttal video (03:26 - 03:32). The temporal flickering and blurry artifacts demonstrate that it does not interpolate well across time or views compared to dynamic NeRF, especially in our large motion and sparse view (9 views) setting.
> * Moreover, as shown in the rebuttal video (02:50 - 02:58), such artifacts are also observed in the STAG4D and DreamGaussian4D (both adopt 4D Gaussian representation) results, even though they sample denser views for SDS loss. This is due to the discrete nature of Gaussian representations, whereas the MLPs in NeRF can learn a more continuous function over time and 3D space.
> * Note that the NeRF-based representation also allows us to render surface normal for normal regularization more conveniently, which is crucial for 4D reconstruction in this ill-posed setting as well as mesh extraction with smooth surfaces.
> * We will add these points in Sec 3.2 in the final draft. We have left a placeholder line there for now to restrict the draft to the page limit.
>
> In light of our point-by-point responses, it would be great if the reviewer could increase the score of our paper and champion it for publication. We thank the reviewer for the feedback and continued support for a quality publication.

---

> ### Comment · Reviewer_p3AT · 2024-11-25
> **Response to Author Rebuttal**
>
> Thanks for the effort of the authors in rebuttal.
> 1) I suggest to illustrate the blending of spatial and temporal attention outputs in pipeline figure for better understanding.
> 2) I watch the supplementary rebuttal video, and find generation results on real-world input video are not very good. I guess although SVD have generalization ability to real-world videos, SV3D may not have, and that's why the performance is not very good. (Also, fintuning on synthetic data will lose some generalization ability of SVD).  But, thanks the authors for testing their model on real-world data.
> 3) I reserve my points about long video generation. The sampling strategy mentioned in rebuttal makes it hard to preserve  the appearance of the object in very long video.
>
> Despite of above concerns, this work is encouraging for building a 4D diffusion model, and I'll keep my positive rate.

---

> ### Author Response · Authors · 2024-11-25
>
> We thank the reviewer for the additional feedback:
> * We followed the reviewer's suggestion and updated the pipeline figure (Figure 2) in the revised paper PDF.
> * Despite the slightly worse results on real-world videos compared to synthetic data, we want to emphasize that SV4D produces higher-quality outputs than the prior works on both novel-view video synthesis (original supplementary video 04:14 rows 1 and 3) and 4D generation (original supplementary video 05:26 rows 2 and 4). Note that we also show additional results on several in-the-wild videos in our rebuttal video (01:30 - 02:03) to demonstrate the generalization capability. Considering that most other 4D generation works only show results on synthetic videos, we believe that SV4D serves as a crucial first step towards generalizable video-to-4D generation and will encourage future research in this direction.
> * We acknowledge SV4D’s limitation of generating really long videos while maintaining strong temporal correspondence, and will rephrase the claim accordingly. We would also like to point out that such limitation is shared among all existing 4D generation methods, whereas our novel sampling strategy can mitigate the issue more effectively than video interpolation-based approaches, as shown in Figure 8 in our manuscript.
>
> We appreciate the constructive comments by the reviewer and acknowledge the limitations of SV4D. Despite the challenges and shortcomings, we believe that the proposed SV4D model and extensive evaluation provide great contributions to the community. We thank the reviewer for their positive encouragement for building a 4D diffusion model, and hope that the reviewer can further increase their score if possible.

---

### Official Review · Reviewer_HfhR · 2024-11-04

**Soundness:** 2
**Presentation:** 3
**Contribution:** 2
**Rating:** 5
**Confidence:** 4

**Summary:**

Stable Video 4D (SV4D) introduces a unified latent video diffusion model for generating multi-frame and multi-view consistent 4D content. Unlike previous methods that use separate models for video generation and novel view synthesis, SV4D simultaneously generates novel view videos from a monocular reference video to ensure temporal consistency. These generated videos are then used to optimize a dynamic NeRF without SDS-based optimization. Trained on a curated dynamic 3D object dataset from Objaverse, SV4D demonstrates sota performance in novel-view video synthesis and 4D generation.

**Strengths:**

1. Performance: The proposed method SV4D achieves state-of-the-art results. The experiments well validate the effectiveness of the proposed methods.

2. Clarity: The paper is well-written and clearly structured, making the methodology and results easy to understand and follow.

3. Technical Novelty: This paper presents Stable Video 4D (SV4D), a latent video diffusion model that simultaneously handles multi-frame video generation and multi-view synthesis for dynamic 3D objects. Unlike previous approaches that utilize separate generative models for each task, SV4D ensures temporal and spatial consistency by generating novel view videos directly from a monocular reference video.

**Weaknesses:**

1. For the claim in Section 3.2, line300, "we observe that this dynamic NeRF representation produces better 4D results compared to other representations such as 4D Gaussian Splatting (Wu et al., 2024), which suffers from flickering artifacts and does not interpolate well across time or views.", do you have any experimental results to support this?

2. The design of SV4D is closely aligned with that of EG4D, as both utilize SVD and SV3D priors. SV4D uses dynamic NeRF and EG4D uses dynamic GS. This significant overlap raises concerns regarding the distinctiveness and novelty of SV4D. To clearly differentiate these two works, the authors should introduce new innovative elements or highlight unique aspects.

**Questions:**

Please refer to weaknesses.

---

> ### Author Response · Authors · 2024-11-21
>
> We thank the reviewer for the valuable feedback. According to the suggestions, we have **updated our paper PDF** and **included a rebuttal video** in the Supplementary Material. We also address the additional comments and questions below.
>
> **W1: Comparison with 4D Gaussians**
> * We show additional results using a 4D Gaussian representation in the rebuttal video (03:26 - 03:32). The temporal flickering and blurry artifacts demonstrate that 4D Gaussian representation does not interpolate well across time or views compared to dynamic NeRF, especially in our large motion and sparse view (9 views) setting.
> * As shown in the rebuttal video (02:50 - 02:58), such artifacts are also observed in the results of STAG4D and DreamGaussian4D (both adopt 4D Gaussian representation), even though they sample denser views for SDS loss. This is due to the discrete nature of Gaussian representations, whereas the MLPs in NeRF can learn a more continuous function over time and 3D space.
> * Note that the NeRF-based representation also allows us to render surface normal for normal regularization more conveniently, but 4D Gaussian representation does not. This aspect is crucial for 4D reconstruction in this ill-posed setting, as well as in mesh extraction with smooth surfaces.
> * We will add these points in Sec 3.2 in the final draft. We have left a placeholder line there for now to restrict the draft to the page limit.
>
> **W2: Differences between EG4D and SV4D**
> * As mentioned in Sec 2: lines 141-144, the key difference between EG4D and SV4D is that EG4D simply combines the SVD and SV3D priors in an **inference-only framework**, whereas SV4D **trains** a 4D (multi-view video) diffusion model.
> * Specifically, EG4D generates multi-view videos by first sampling a monocular video via SVD, then temporally blending the spatial keys and values in SV3D’s attention modules **at inference**, during multi-view generation.
> * Similar **inference-only** methods are also explored in Diffusion^2 and STAG4D, as mentioned in Sec 4: lines 362-364. Diffusion^2 blends the image latents denoised by SVD and SV3D during sampling. STAG4D temporally blends the attention keys and values in Zero123++ during novel-view synthesis. These simple blending strategies to utilize separately-trained models often cause blurry details or poor spatio-temporal consistency, and thus can only handle small object motion.
> * As EG4D’s code is not public, we show the results of Diffusion^2 and STAG4D in our rebuttal video (02:03 - 02:40) for reference.
> * On the contrary, SV4D proposes a unified 4D model which jointly learns spatial and temporal consistency from 4D data, while reusing the rich priors from SVD and SV3D via attention weights initialization. We show both visually and quantitatively that SV4D produces higher-fidelity and more consistent details in the synthesized novel-view videos, especially in the presence of large motion.
> * Please also note that EG4D additionally relies on a third model (SDXL-Turbo) to refine the 4D assets, whereas our 4D optimization only leverages the videos generated by SV4D.
> * We have highlighted these aspects about SV4D in the revised paper in Sec 2.
>
> In light of our point-by-point responses, it would be great if the reviewer could increase the score of our paper and champion it for publication. We thank the reviewer for the feedback and continued support for a quality publication.

---

> > ### Author Response · Authors · 2024-11-25
> >
> > We thank the reviewer again for the feedback on our paper!
> > As the discussion deadline (Nov. 26) is approaching, we were wondering whether the reviewer had the chance to look at our response and whether there is anything else the reviewer would like us to clarify. We sincerely hope that our response has addressed the concerns, and if so, we would be grateful if the reviewer could consider increasing the score accordingly.
> >
> > Best,
> > SV4D authors

---

> > > ### Author Response · Authors · 2024-11-27
> > >
> > > We thank Reviewer `HfhR` once again for taking the time to review our work. We wanted to kindly check if Reviewer `HfhR` had an opportunity to review our response and if there are any additional clarifications needed. We sincerely hope our response has addressed the concerns and would greatly appreciate it if Reviewer `HfhR` could consider increasing the score accordingly.
> > >
> > > Best,
> > > SV4D authors

---

> ### Author Response · Authors · 2024-12-02
> **Awaiting Response from Reviewer HfhR (Final Day Remaining)**
>
> We sincerely thank Reviewer `HfhR` for taking the time to review our work. With just **one day remaining** until the deadline, we kindly request feedback on our response. If any part of our explanation is unclear, please let us know. We genuinely hope our response has addressed the concerns and would greatly appreciate it if the reviewer could consider raising the score accordingly.
>
> Best,
> SV4D authors

---

### Author Response · Authors · 2024-11-21

We thank the reviewers for their time and effort on providing the valuable feedback. We are glad to see that
* The design of SV4D is recognized as “**technically novel**” (HfhR)
* Our work “**have sufficient value to the community**” (p3AT, EfLA)
* The experiments “**well validate the effectiveness**" (HfhR, p3AT) and “**prove the generalization ability**” (p3AT)
* The proposed "**mix-sampling strategy is interesting**” (EfLA)
* The paper is “**well-organized and easy to follow**” (HfhR, EfLA)

According to the feedback, we have **updated our paper PDF** and **included a rebuttal video in the Supplementary Material** for the reviewers’ perusal. Changes are marked in yellow in the updated version of the paper and appendix.

---

### Meta-Review · Area_Chair_WKAa · 2024-12-17

**Metareview:**

This paper receives ratings of 5,6,8,6. The AC follows the suggestions of the reviewers to accept the paper. The paper is well-written and is easy to follow. The proposed method of mix-sampling strategy, and the use of view attention and frame attention blocks are effective in solving the task. The attempt to train a multi-view video diffusion model for 4D generation is encouraging and is valuable to the community. The paper shows extensive comparisons to previous methods. Although there's a reviewer who gave a negative rating of 5, the AC thinks that the weaknesses pointed out by the reviewer are well-addressed by the authors in the rebuttal and are not significant to reject the paper. Furthermore, the reviewer also mentioned the performance, clarity and technical novelty as the strengths of the paper. These strengths are well-aligned with those pointed out by the other reviewers. Overall, all the weaknesses pointed out by the reviewers are also well-addressed by the authors, and the AC advises the authors to include these updates in their final camera-ready version.

**Additional Comments On Reviewer Discussion:**

The weaknesses mentioned by the reviewers are mostly clarifications. These are well-addressed by the authors during the rebuttal and discussion phases.

---

### Decision · Program_Chairs · 2025-01-22

Accept (Poster)